**Measurement report: Large contribution of biomass burning and aqueous-phase processes to the wintertime secondary organic aerosol formation in Xi'an, Northwest China**

Jing Duan[1], Ru-Jin Huang[1,2], Yifang Gu[1,2], Chunshui Lin[1], Haobin Zhong[1,2], Wei Xu[1], Quan Liu[3], Yan You[4], Jurgita Ovadnevaite[5], Darius Ceburnis[5], Thorsten Hoffmann[6], Colin O'Dowd[5]

[1]State Key Laboratory of Loess and Quaternary Geology (SKLLQG), CAS Center for Excellence in Quaternary Science and Global Change, Institute of Earth Environment, Chinese Academy of Sciences, Xi'an 710061, China

[2]University of Chinese Academy of Sciences, Beijing 100049, China

[3]China State Key Laboratory of Severe Weather & Key Laboratory of Atmospheric Chemistry of CMA, Chinese Academy of Meteorological Sciences, Beijing 100081, China

[4]National Observation and Research Station of Coastal Ecological Environments in Macao, Macao Environmental Research Institute, Macau University of Science and Technology, Macao SAR 999078, China

[5]School of Physics and Centre for Climate and Air Pollution Studies, Ryan Institute, National University of Ireland Galway, University Road, Galway, H91CF50, Ireland

[6]Department of Chemistry, Johannes Gutenberg University Mainz, Duesbergweg 10−14, Mainz 55128, Germany

**Correspondence**: Ru-Jin Huang (rujin.huang@ieecas.cn) or Quan Liu (liuq@cma.gov.cn)

**Abstract**

Secondary organic aerosol (SOA) plays an important role in particulate air pollution, but its formation mechanism is still not fully understood. The chemical composition of non-refractory particulate matter with a diameter ≤ 2.5 μm (NR-PM$_{2.5}$), OA sources, and SOA formation mechanisms were investigated in urban Xi'an during winter 2018. The fractional contribution of SOA to total OA mass (58%) was larger than primary OA (POA, 42%). A biomass burning-influenced oxygenated OA (OOA-BB) was resolved in urban Xi'an, which was formed from the photochemical oxidation and aging of biomass burning OA (BBOA). The formation of OOA-BB was more favorable in the days with larger OA fraction and higher BBOA concentration. In comparison, the aqueous-phase processed oxygenated OA (aq-OOA) was more dependent on secondary inorganic aerosol (SIA) content and aerosol liquid water content (ALWC), and increased largely to 50% of OA during SIA-enhanced periods. Further Van Krevelen (VK) diagram analysis suggests the increased aq-OOA contributions during SIA-enhanced periods were likely from alcohol or peroxide addition in the OA aqueous-phase oxidation processes.

**1 Introduction**

Particulate matter with a diameter ≤ 2.5 μm (PM$_{2.5}$) in the atmosphere has become an important environmental problem for climate, visibility and human health, especially in China with rapid

industrialization, urbanization and population expansion (Huang et al., 2014; Lelieveld et al., 2015; Peng et al., 2016; An et al., 2019). Most megacities in China are frequently plagued by severe particulate pollution in recent years, attracting extensive attention and research on its composition characteristics and formation mechanisms (Guo et al., 2014; Hu et al., 2013, 2016; Li et al., 2017; Tong et al., 2017; Sun et al., 2016, 2018). The haze pollution occurs more frequently in winter with unfavorable meteorological conditions, and myriad variables, such as complex emission sources, pollutant lifetimes, and atmospheric reactions (Sun et al., 2013, 2014; Elser et al., 2016; Hu et al., 2016; An et al., 2019; Kuang et al., 2020).

Fine particles can be either emitted directly from primary sources that refers to as primary aerosol, or produced in the atmosphere through gas-to-particle conversion or aging of primary aerosol, which refers to as secondary aerosol (Jimenez et al., 2009; Liu et al., 2010; Xu et al., 2017). Numerous studies have elucidated the increasing importance of secondary aerosol in haze pollution (Sun et al., 2016; Huang et al., 2014, 2019; An et al., 2019; Duan et al., 2020). However, the formation and evolution of secondary aerosol, especially secondary organic aerosol (SOA), is still not well understood and becoming a critical concern for air pollution research (Gilardoni et al., 2016; Xu et al., 2017; Kuang et al., 2020; Zhang, H. et al., 2021; Li, J. et al., 2022; Lv et al., 2022). Variable precursors, complex transformation and aging chemistry of SOA lead to insufficient cognition of its formation and uncertainty in model simulation (Shrivastava et al., 2017).

Field studies based on aerosol mass spectrometer (AMS) combined with OA source apportionment techniques (Paatero, 1999; DeCarlo et al., 2006; Canonaco et al., 2013) have been conducted in China to resolve SOA sources and investigate its formation and evolution mechanisms (Hu et al., 2013, 2016; Sun et al., 2016; Xu et al., 2017, 2019). Gas-phase photochemical oxidation has been considered as a major pathway of SOA formation in a number of studies, according to the correlation between SOA and odd oxygen, which defined as Ox (Ox = $O_3$+$NO_2$) (Sun et al., 2014; Elser et al., 2016; Hu et al., 2016). However, recent studies also revealed the important contribution of aqueous-phase chemistry to SOA formation, which is also missing in SOA simulation and difficult to identify (Guo et al., 2014; Sun et al., 2016; Xu et al., 2017, 2019; Huang et al., 2020; Li et al., 2021). For example, Sun et al. (2016) resolved an aqueous-phase-processed SOA (aq-OOA) which significantly affected OA oxidation state in high RH conditions (>50%). The results of Wang et al (2017) and Xu et al. (2017) indicated that aqueous-phase chemistry played a dominant role in the formation of more-oxidized-oxygenated OA (MO-OOA). Kuang et al. (2020) further resolved the contribution of photochemical aqueous-phase chemistry in wintertime haze pollution which induced the rapid formation of SOA in the daytime.

As the largest city of the Guanzhong basin, one of the top three regions in China's air cleaning campaign, Xi'an has suffered serious particulate pollution in recent years due to the rapid urbanization, while research on aerosol composition and SOA formation mechanisms are still limited in the region (Elser et al., 2016; Zhong et al., 2020; Duan et al., 2021). Elser et al (2016) analyzed the chemical composition and OA sources of $PM_{2.5}$ during the heavy pollution period of 2013 in Xi'an using a high resolution

AMS (HR-AMS), and found the contribution of SOA increased during extreme haze events, but the SOA formation mechanism and OA oxidation state during haze pollution were not well analyzed. As multiple control measures have been implemented in Xi'an, such as the 13th five-year energy conservation and emission reduction plan (Wan et al., 2022), and motor vehicle restrictions, it is expected that aerosol composition and sources have largely varied in recent years, while direct elucidation and characterization are lack. Recent studies showed that biomass burning and secondary formation dominated OA concentration in Xi'an, which in total contributed >50% of total OA in both autumn and winter (Zhong et al., 2020). In addition, Xiao et al. (2020) reported that biomass burning sources, especially residential biofuel, can contribute to increased urban $NH_3$ emissions. Several studies also indicated that biomass burning is an important source of light absorption components in Xi'an (Zhang et al., 2020; Yuan et al., 2021; Zhang, T. et al., 2021; Li, X. et al., 2022). Wu et al. (2018) revealed that simultaneously elevated RH and anthropogenic secondary inorganic aerosol resulted in an abundant ALWC, which can further facilitate the formation of heavy haze. Zhong et al. (2020) indicated that OOA formation was most likely dominated by aqueous-phase processes when Ox was <35 ppb in autumn and winter Xi'an, and Duan et al. (2021) found that persistently high RH/ALWC was the driving factor of aq-OOA formation in summer Xi'an, and the increasing trend of aq-OOA was much consistent with that of nitrate. These studies indicated the importance of biomass burning as well as aqueous-phase reactions in Xi'an, which need further elucidation.

In this study, $PM_{2.5}$ composition was measured during the heating season of 2018 in Xi'an using a soot particle long-time-of-flight AMS (SP-LToF-AMS). Chemical composition and OA sources were analyzed and compared with those resolved in Elser et al (2016), in order to elucidate the aerosol variation in recent years due to emission controls. Meanwhile, the SOA formation mechanisms and its contribution to haze event were investigated and compared with those in the summer of 2019 (Duan et al., 2021).

## 2 Experimental

### 2.1 Sampling

The winter campaign was conducted from 4[th] December 2018 to 15[th] March 2019 at the campus of the Institute of Earth Environment, Chinese Academy of Sciences (34°23′N, 108°89′, 12 m above the ground level) in downtown Xi'an with surrounding residential, commercial, and traffic areas (Duan et al., 2021).

A SP-LToF-AMS (Aerodyne Research Inc.) was deployed for the online characterization of $PM_{2.5}$ with a time resolution of 1 min. The detailed instrument description could be found in Onasch et al. (2012) and a similar operation was conducted as that in Duan et al. (2021). The contribution of black carbon (BC) was not considered, and only NR-$PM_{2.5}$ composition, including organics (OA), nitrate ($NO_3^-$), sulfate ($SO_4^{2-}$), ammonium ($NH_4^+$), and chloride ($Cl^-$) were analyzed. Briefly, ambient air was sampled into the room at a flow rate of 5 L min$^{-1}$. After being dried by a Nafion dryer (MD-700-24S, Perma

Pure, Inc.), the ambient aerosol was focused into a particle beam using an $PM_{2.5}$ aerodynamic lens, and was sub-sampled into the SP-LToF-AMS at a flow rate of ~ 0.1 L min$^{-1}$. The particle beam was then vaporized upon impacting the heated tungsten surface (~ 600 °C), and ionized by electron ionization (70 eV) to produce positive fragments, which were detected and analyzed by the LToF mass spectrometer. The ionization efficiency (IE) as well as relative ionization efficiency (RIE) calibrations were conducted during the campaign, using the 350 nm (Dm) ammonium nitrate ($NH_4NO_3$) and ammonium sulfate (($NH_4)_2SO_4$) particles (Jimenez et al., 2003). Meanwhile, gases species including CO, $NO_2$, $O_3$ and $SO_2$ were measured using a Thermo Scientific Model 48i carbon monoxide analyzer, a Thermo Scientific Model 42i NO–$NO_2$–$NO_x$ analyzer, a Thermo Scientific Model 49i ozone analyzer, and an Ecotech EC 9850 sulfur dioxide analyzer, respectively. The meteorological parameters including relative humidity (RH), temperature, wind speed, and wind direction were measured by an automatic weather station (MAWS201, Vaisala, Vantaa, Finland) and a wind sensor (Vaisala Model QMW101-M2), respectively.

**2.2 Data analysis**

The SQUIRREL (version 1.61D) and PIKA (1.21D) coded in Igor Pro 6.37 (WaveMetrics) were used to analyze the SP-LToF-AMS data. Standard RIEs of 1.4, 1.1 and 1.3 were used for organics, nitrate and chloride, respectively, while experimentally determined RIEs of 3.7 and 1.3 were used for ammonium and sulfate, respectively. Meanwhile, the composition-dependent collection efficiency (CDCE) was used to calibrate and compensate for the incomplete detection due to particle bounce (Middlebrook et al., 2012). Note RH was not considered in the CDCE calculations as a Nafion dryer was used and the RH effects on collection efficiency were much reduced. The elemental ratios including oxygen-to-carbon (O/C), organic mass-to-organic carbon (OM/OC) and hydrogen-to-carbon (H/C) were also analyzed for the high-resolution OA mass spectra based on the Improved Ambient (I-A) method (Canagaratna et al., 2015). Meanwhile, the data and error matrices of high-resolution OA mass spectra for m/z 12-120 were preprocessed, and OA source apportionment was performed using Positive Matrix Factorization (PMF) and multilinear engine (ME-2) in Igor Pro (Paatero, 1999), as conducted in Duan et al (2021).

In addition, the aerosol liquid water content (ALWC) was also calculated based on the ISORROPIA-II model, using inorganic aerosol composition ($NH_4^+$, $SO_4^{2-}$, $NO_3^-$, $Cl^-$) combined with ambient temperature and RH as input data (Fountoukis and Nenes, 2007). The simulation was run in "metastable" mode in which all components are assumed to be deliquescent and no solid matter is present. The thermodynamic equilibrium and phase state of those inorganic species were then simulated and the ALWC was resolved.

**3 Results and discussion**

**3.1 Overview of NR-PM$_{2.5}$ composition and OA sources in winter Xi'an**

During the winter of 2018 in Xi'an, NR-PM$_{2.5}$ concentration varied from 5.9 μg m$^{-3}$ to 205.6 μg m$^{-3}$,

with an average of $68.0 \pm 42.8$ μg m$^{-3}$ (see Fig. 1 and Table S1, note that all the values throughout the results and discussion are the arithmetic means and standard deviations of the per-minute samples over the campaign or specified sub-period). This average concentration was higher than that measured in the summer of 2019 in Xi'an ($22.3 \pm 11.7$ μg m$^{-3}$, Duan et al., 2021), due to the increase of source emissions in winter than in summer which was also observed in other cities (Sun et al., 2015; Xu et al., 2014,

2016). Meanwhile, the average NR-PM$_{2.5}$ concentration observed in our study was much lower than those observed in the winter of 2013 in Xi'an ($125.0 \pm 99.0$ μg m$^{-3}$ during reference days and $498.0 \pm 146.0$ μg m$^{-3}$ during haze days, respectively) (Elser et al., 2016), pointing to an improved air quality. However, haze events with NR-PM$_{2.5}$ concentrations higher than 100 μg m$^{-3}$ were still observed frequently during the campaign, indicating some overlooked pollution sources or atmospheric formation

pathways which require further attention. As for the chemical composition, OA constituted a dominant fraction of 54% in total NR-PM$_{2.5}$ mass, lower than that observed in summer Xi'an (63%). Nitrate contributed 20% to total NR-PM$_{2.5}$ mass, followed by sulfate (13%), ammonium (10%), and chloride (3%). The higher contribution of nitrate than sulfate was opposite to that in summer with higher contribution from sulfate (17%) than nitrate (12%), suggesting the increased formation and contribution

of nitrate in winter pollution, likely due to the much lower temperature in winter which facilitated the transformation of nitrate from gas-phase to particle-phase (Duan et al., 2021). Meanwhile, the contribution of nitrate in our campaign was also higher than that observed in Xi'an during the winter of 2013 (by 13% during haze days and by 10% during reference days, respectively) (Elser et al., 2016), suggesting the increasing importance of nitrate pollution over sulfate pollution in recent years,

consistent with the interannual evolution trend of nitrate observed in Beijing (Xu et al, 2019).

     A continuous and large increase of secondary inorganic aerosol (SIA, nitrate + sulfate + ammonium) was observed during two periods, including period 1 from 2018/12/30 0:00 to 2019/1/15 6:00 (SIA-enhanced period 1, SIA_P1) and period 2 from 2019/2/7 0:00 to 2019/3/4 23:00 (SIA-enhanced period 2, SIA_P2). The other periods are defined as reference days. During the reference days, OA contributed

a major fraction of 66% to total NR-PM$_{2.5}$ mass, even higher than that during the summer of 2019 in Xi'an (63%) (Duan et al., 2021). In comparison, from reference days to SIA_P1 and SIA_P2, the contribution of OA decreased from 66% to 52% and 44%, respectively, and the contribution of SIA increased from 30% to 45% and 53%, accordingly. Meanwhile, the SIA-enhanced periods were also related to higher PM$_{2.5}$ concentration, which increased from $44.1 \pm 25.5$ μg m$^{-3}$ during reference days to

$131.0 \pm 49.6$ μg m$^{-3}$ during SIA_P1 and $84.9 \pm 30.7$ μg m$^{-3}$ during SIA_P2, suggesting the much important contribution of SIA in the formation of haze pollution in winter Xi'an (Zhong et al., 2020; Zhang et al., 2021). The major difference between SIA-enhanced periods and reference days was the much frequent occurrence of higher relative humidity (RH>60%) and ALWC concentration (ALWC >10 μg m$^{-3}$) during SIA_P1 and SIA_P2 than reference days (Fig. S1). These indicated the

more frequent occurrence of liquid condition during SIA-enhanced periods than reference days. According to previous studies, high RH and liquid phase reactions played important roles in the formation of secondary inorganic aerosol, such as sulfate and nitrate (Sun et al., 2016; Wu et al., 2018).

These indicated that high RH and liquid phase condition may drive the large production of SIA in winter Xi'an (Xu et al., 2019; Duan et al., 2021).

During our measurement, the concentration of ammonium increased from $3.3 \pm 2.2$ µg m$^{-3}$ during reference days to $13.3 \pm 6.5$ µg m$^{-3}$ during SIA_P1 and $10.8 \pm 4.6$ µg m$^{-3}$ during SIA_P2, consistent with the variation trends of sulfate and nitrate, in which sulfate increased from $3.5 \pm 2.8$ µg m$^{-3}$ during reference days to $18.4 \pm 10.2$ µg m$^{-3}$ during SIA_P1 and $14.7 \pm 7.2$ µg m$^{-3}$ during SIA_P2, and nitrate increased from $6.8 \pm 4.9$ µg m$^{-3}$ during reference days to $27.4 \pm 13.4$ µg m$^{-3}$ during SIA_P1 and $19.9 \pm$

$9.3$ µg m$^{-3}$ during SIA_P2. The equivalent molar concentration of ammonium correlated tightly with that of the total of sulfate and nitrate with a slope $\approx 1$ during all the three periods including reference days, SIA_P1, and SIA_P2, suggesting ammonium was mainly neutralized by sulfate and nitrate in winter Xi'an both in reference days and SIA-enhanced periods (Fig. S2).

Specifically, in order to further analyze the relative importance of sulfate and nitrate in haze pollution,
the increase ratio of sulfate or nitrate contribution from reference days to SIA periods was calculated following the equations below:

$$\text{IR}_{\text{sulfate}}=f_{\text{sulfate,SIA}}/f_{\text{sulfate,reference}}$$

$$\text{IR}_{\text{nitrate}}=f_{\text{nitrate,SIA}}/f_{\text{nitrate,reference}}$$

In which the $\text{IR}_{\text{sulfate}}$ and $\text{IR}_{\text{nitrate}}$ refers the increase ratio of sulfate contribution or nitrate contribution
from reference days to SIA periods, respectively. $f_{\text{sulfate,SIA}}$ or $f_{\text{nitrate,SIA}}$ refers the mass fraction of sulfate or nitrate in total PM$_{2.5}$ mass during SIA periods including SIA_P1 and SIA_P2, and $f_{\text{sulfate,reference}}$ or $f_{\text{nitrate,reference}}$ refers the mass fraction of sulfate or nitrate in total PM$_{2.5}$ mass during reference days.

The $\text{IR}_{\text{sulfate}}$ from reference days to SIA_P1 (1.8) and to SIA_P2 (2.1) was higher than those of $\text{IR}_{\text{nitrate}}$ (1.4 from reference days to SIA_P1 and 1.5 from reference days to SIA_P2, respectively). Meanwhile,
the average mass ratio of NO$_3^-$/SO$_4^{2-}$ (Sun et al., 2016) decreased from 1.9 during reference days to 1.5 during SIA_P1 and 1.4 during SIA_P2, respectively. These trends suggested that the increase of sulfate contribution during haze pollution was much obvious than that of nitrate contribution in winter Xi'an, although the absolute concentration of nitrate was higher than sulfate both in reference days and SIA periods. NO$_3^-$/SO$_4^{2-}$ showed an evident decrease as a function of RH at higher NR-PM$_{2.5}$ loading ($> 50$
µg m$^{-3}$) (Fig. S1). Consistently, although both sulfur oxidation ratio (SOR, defined as n[SO$_4^{2-}$]/(n[SO$_4^{2-}$] + n[SO$_2$]), Ji et al., 2018; Chang et al., 2020) and nitrogen oxidation ratio (NOR, defined as n[NO$_3^-$]/(n[NO$_3^-$] + n[NO$_2$]), Ji et al., 2018; Chang et al., 2020) increased with RH, SOR increased from 0.10-0.20 at RH $< 40\%$ to 0.33-0.63 at RH $> 60\%$, which was more efficient than the increase of NOR (from 0.07-0.10 at RH $< 40\%$ to 0.18-0.30 at RH $> 60\%$) (Fig. S3). These results suggested that
high RH is favorable in sulfate formation than nitrate formation especially in haze pollution in winter Xi'an.

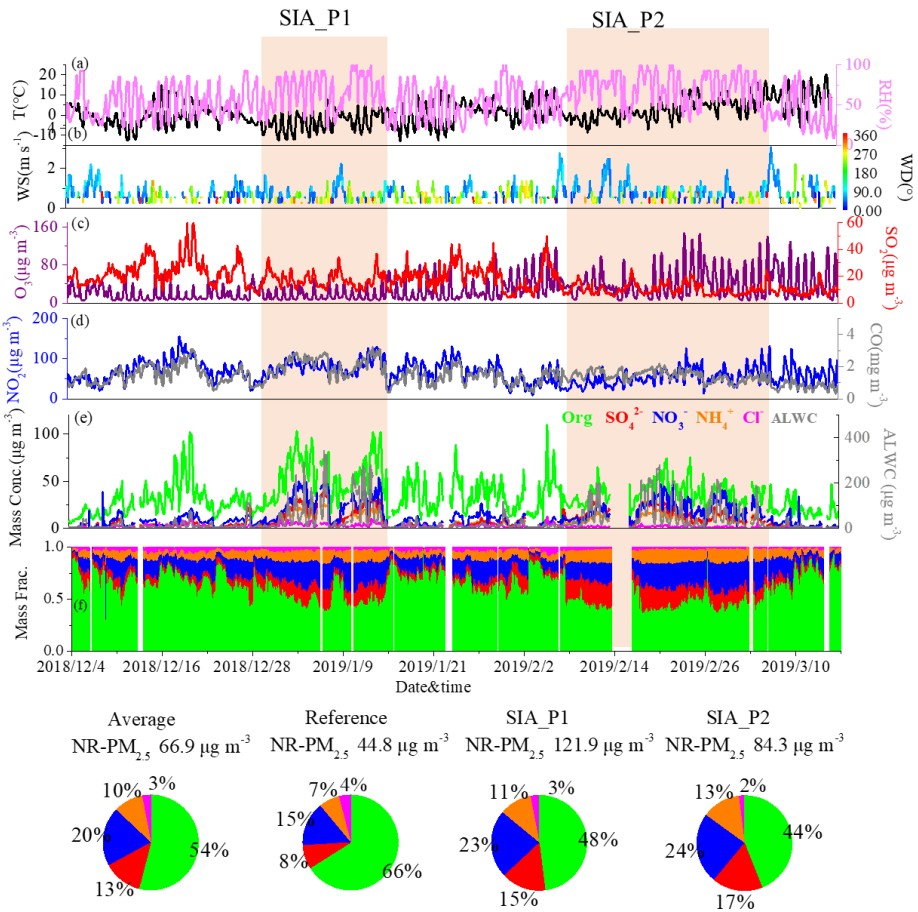

**Fig. 1** Time series of meteorology parameters (relative humidity (RH), temperature (T), wind speed (WS), and wind direction (WD) (a, b); gases species ($SO_2$, $O_3$, $NO_2$ and CO) (c, d); and NR-PM$_{2.5}$ composition as well as the aerosol liquid water content (ALWC) (e, f) in the winter of 2018 in Xi'an. The average composition of NR-PM$_{2.5}$ for the entire winter campaign, as well as reference days and SIA-enhanced periods (SIA_P1 and SIA_P2) are also shown.

Six OA sources were resolved, including a hydrocarbon-like OA (HOA), a cooking OA (COA), a biomass burning OA (BBOA), a coal combustion OA (CCOA), a biomass burning influenced-oxygenated OA (OOA-BB), and an aqueous phase processed-oxygenated OA (aq-OOA) (Fig. 2, the OA source apportionment was detailed in the supplement). POA including HOA, COA, CCOA and BBOA in total contributed 42% to OA mass. HOA contributed 8% ($3.0 \pm 3.9$ μg m$^{-3}$) to the total OA mass (Fig. 2). This contribution was lower than that observed in winter 2013 (18%, $23.0 \pm 27.0$ μg m$^{-3}$ in reference days and 16%, $49.0 \pm 41.0$ μg m$^{-3}$ in extreme haze, respectively) (Elser et al., 2016), which may be related to the better traffic control in recent years in urban Xi'an. COA on average contributed 13% ($4.8 \pm 4.2$ μg m$^{-3}$) to total OA, which was higher than that observed during winter 2013 (9%, $15.8 \pm 8.7$ μg m$^{-3}$ in reference days and 4%, $33.0 \pm 16.0$ μg m$^{-3}$ in extreme haze, respectively) in Xi'an (Elser, et al., 2016). CCOA on average contributed 9% ($3.2 \pm 2.5$ μg m$^{-3}$) to total OA in this winter campaign, consistent with that observed in the winter of 2013 (14%, $5.7 \pm 4.1$ μg m$^{-3}$ in reference days and 6%, $7.7 \pm 8.0$ μg m$^{-3}$ in extreme haze, respectively) (Elser et al., 2016). In comparison, BBOA was more

significant contributor than CCOA, and on average accounted for 12% (4.3 ± 5.9 μg m⁻³) of total OA mass. However, this contribution was much lower than that observed in the winter of 2013 in Xi'an (42%, 22.0 ± 20.0 μg m⁻³ in reference days and 43%, 67.0 ± 40.0 μg m⁻³ in extreme haze, respectively) (Elser et al., 2016), suggesting the reduction of BBOA emissions in recent years in Xi'an and surrounding areas. SOA contributed a higher fraction of 58% (21.8 ± 7.4 μg m⁻³) than POA to total OA, with OOA-BB and aq-OOA accounting for 24% and 34% of OA mass, respectively. The contribution of SOA was much higher than that observed in the winter of 2013 in Xi'an (16%, 5.4 ± 8.9 μg m⁻³ in reference days and 31%, 47.0 ± 12.0 μg m⁻³ in haze days, respectively) (Elser et al., 2016).

As discussed above, the SIA-enhanced periods were usually related to haze pollution with higher NR-PM₂.₅ mass. OA composition between reference days and SIA-enhanced periods was further compared, in order to better understand the OA evolution during haze pollution in Xi'an (Fig. S9). From reference days to SIA_P1, the total mass of OA increased from 28.7 ± 16.4 μg m⁻³ to 68.0 ± 20.7 μg m⁻³ (Table S1). Both POAs and SOAs concentrations increased, with the aq-OOA increasing the most from 4.9 ± 3.7 μg m⁻³ to 26.2 ± 14.6 μg m⁻³. As a result, the O/C ratio of the bulk OA increased from 0.41 ± 0.10 during reference days to 0.52 ± 0.10 during SIA_P1, suggesting the enhanced OA oxidation state during SIA_P1. In comparison, the total mass of OA (37.7 ± 11.7 μg m⁻³) during SIA_P2 was higher than that during reference days, while lower than that during SIA_P1. The mass concentrations of POAs and OOA-BB were lower than those during both reference days and SIA_P1, and the increase of the total OA mass from reference days to SIA_P2 was dominantly ascribed to the dramatic increase of aq-OOA from 4.9 ± 3.7 μg m⁻³ to 22.7 ± 10.7 μg m⁻³, similar with that from reference days to SIA_P1. As a result, the O/C ratio of total OA during SIA_P2 was further enhanced to 0.67 ± 0.11, much higher than those during reference days and SIA_P1.

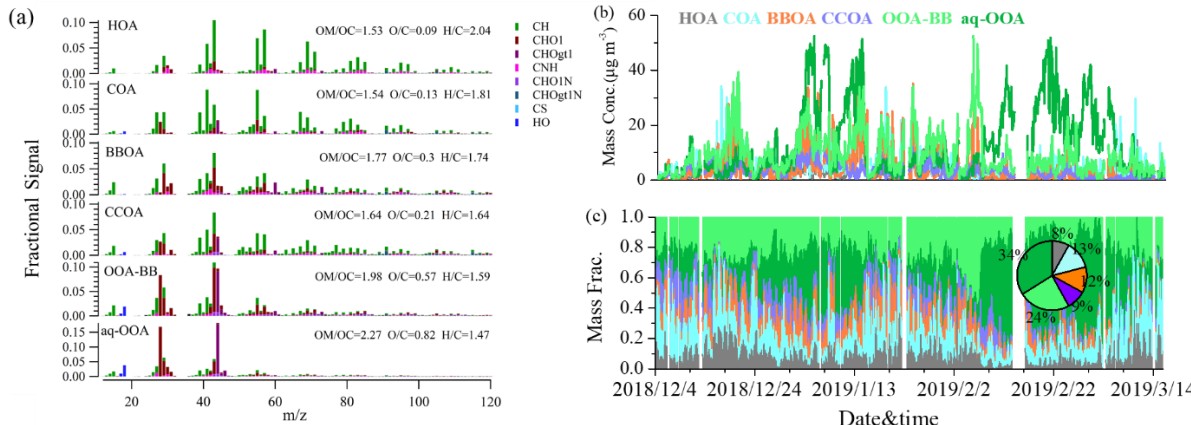

**Fig. 2** Mass spectra of OA sources (a), and time series of concentration (b) and fraction (c) of each OA source in total OA mass during the winter campaign. The average composition of OA sources for the entire observation are also shown as pie chart in figure c.

### 3.2 OOA-BB dependence on BBOA and photochemical oxidation

We further analyzed the evolution and formation mechanism of OOA-BB, and found that the time

variation of OOA-BB correlated well with that of BBOA ($R^2 = 0.59$) (Fig. S7), with peaks of m/z 60 ($C_2H_4O_2^+$) and m/z 73 ($C_3H_5O_2^+$) in the mass spectrum of OOA-BB (Fig. 2), indicating the possible influence of BBOA source on the formation of OOA-BB. Note that although moderate correlation was observed between the time series of OOA-BB and BBOA, lags and differences between their time series were observed, suggesting the atmospheric aging under environmental conditions.

The fragment ions of m/z 60 ($C_2H_4O_2^+$) and m/z 73 ($C_3H_5O_2^+$) generated from the pyrolysis of cellulose such as levoglucosan and mannosan were considered as good tracers of BBOA (Alfarra et al., 2007; Cubison et al., 2011). Fresh BBOA usually exhibits the highest content of m/z 60 ($C_2H_4O_2^+$) and m/z 73 ($C_3H_5O_2^+$), which will decrease due to oxidation reaction and degradation during atmospheric aging. At the same time, oxygenated fragments such as m/z 44 ($CO_2^+$) will increase during atmospheric aging (Cubison et al., 2011; Paglione et al., 2020). The correlation and evolution of $f_{60}$ (the fraction of m/z 60 in the total signal of the OA mass spectrum) and $f_{44}$ (the fraction of m/z 44 in the total signal of the OA mass spectrum) is usually used to analyze the influence of BBOA on SOA and their evolution processes (Cubison et al., 2011). As $PM_{2.5}$ was measured in our campaign, in order to further analyze the influence of BBOA on SOA formation in Xi'an, OA sources in $PM_{2.5}$ resolved using AMS were compared. Fig. 3a displays plots of $f_{44}$ ($f_{CO2+}$) vs. $f_{60}$ ($f_{C2H4O2+}$) of BBOA and SOA sources resolved in the winter of 2018 (this campaign), the winter of 2013 (Elser et a., 2016) and the summer of 2019 (Duan et al., 2021). According to Cubison et al. (2010), $f_{60}$=0.003 ± 0.002 represented the threshold of BB influence. SOA sources have a $f_{60}$ higher than 0.005 suggested the influence from BBOA, while $f_{60}$ < 0.003 suggested a secondary source having no influence from BBOA, and sources from fresh biomass burning emission usually have high $f_{60}$ and low $f_{44}$. As shown in Fig. 3, BBOA factor resolved in the winter of 2018 and 2013 were both located in the fresh BBOA region with higher $f_{C2H4O2+}$ (0.024 and 0.021, respectively, which were both higher than 0.005) and lower $f_{CO2+}$, suggesting they were fresh BBOA emissions. OOA-BB (Paglione et al., 2020) resolved in the winter of 2018 was characterized by a $f_{C2H4O2+}$ value of 0.08 and a $f_{CO2+}$ value of 0.13, which was located in the BB-influenced region, indicating the OOA-BB resolved in the winter of 2018 was largely influenced by BBOA emission. In comparison, the aq-OOA (Sun et al., 2016) resolved in the winter of 2018, OOA resolved in the winter of 2013, as well as the three SOA sources (LO-OOA, MO-OOA, aq-OOA) resolved in the summer of 2019 all showed higher $f_{CO2+}$ and lower $f_{C2H4O2+}$ (< 0.005), and were located in the non-BB influenced region, suggesting that these SOA were formed from other processes independent on BBOA source. In addition, in order to further compare the BBOA influence on SOA between different regions, $f_{44}$ vs. $f_{60}$ for BBOA and SOA resolved in $PM_1$-OA from previous studies were also compared (see Fig. 3b, note that $f_{44}$ and $f_{60}$ values are not available in other group papers, only those resolved in our previous studies are summarized here). In most of studies, BBOA is located in the fresh BBOA region, except the BBOA resolved in the winter of 2012 in Xi'an (Zhong et al., 2020). Meanwhile, most of the SOAs were located in the non-BB influenced region, except the OOA resolved in the winter of 2012 (Zhong et al., 2020) which showed a higher $f_{44}$ of 0.17 and a higher $f_{60}$ of 0.09 (>0.05). This further indicated the influence from biomass burning on SOA formation in winter Xi'an. In comparison, the MO-OOA resolved in Baoji and the

OOA resolved in Shijiazhuang also showed minor influence from BBOA, which are located in the edge of the aged-BB region (Wang et al., 2017; Huang et al., 2019).

In order to further explain the possible pathway of OOA-BB formation and influence from BBOA in the 2018 winter campaign, the evolution of BBOA into OOA-BB was further analyzed using the van Krevelen (VK) diagram of O/C vs. H/C ratios, which is typically used to investigate the OA evolution during field and laboratory experiments (Heald et al., 2010; Ng et al., 2011b). As shown in Fig. 3c, the slope of the line that links BBOA to OOA-BB is -0.59, between −0.5 and −1, suggesting that OOA-BB was likely formed from BBOA through the formation of carboxylic acid moieties (Ng et al., 2011b; Paglione et al., 2020). Meanwhile, in our study, the concentration of OOA-BB positively increased as the Ox increased, suggesting the importance of photochemical oxidation processes (Fig. 3d). Meanwhile, the formation of OOA-BB was also enhanced under higher BBOA concentration conditions, confirming that OOA-BB was formed from the aging of BBOA.

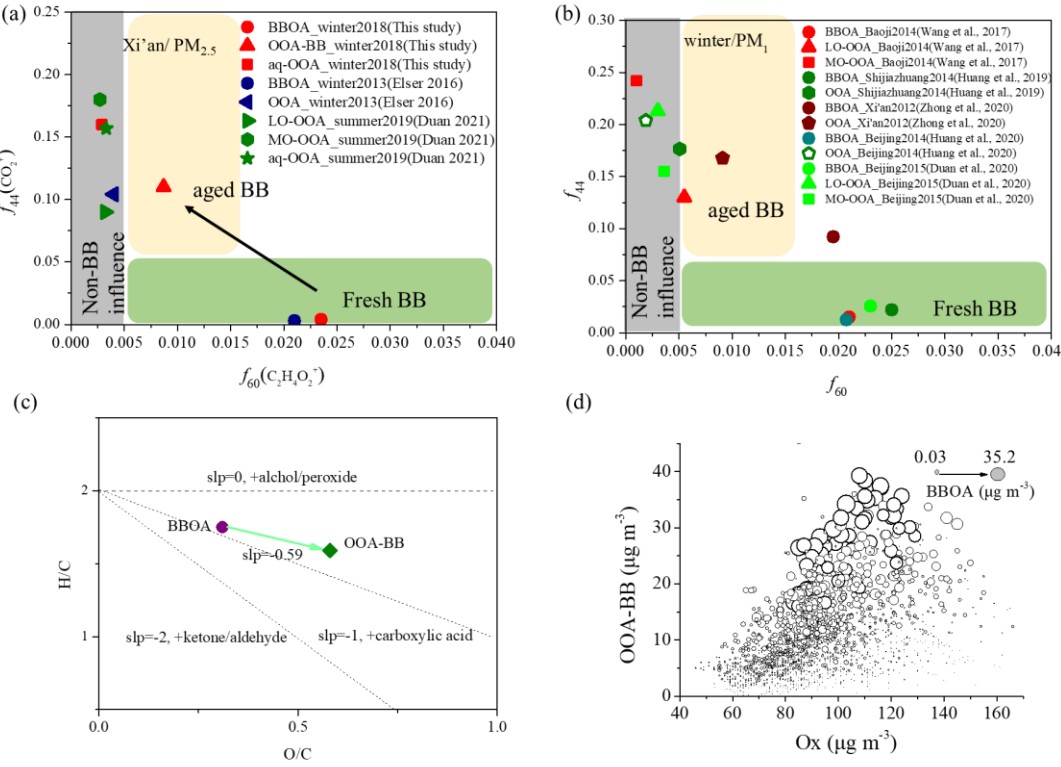

**Fig. 3** Plots of $f_{44}$ vs. $f_{60}$ for BBOAs and SOAs resolved in OA of $PM_{2.5}$ in Xi'an (a); plots of $f_{44}$ vs. $f_{60}$ for BBOAs and SOAs resolved in OA of $PM_1$ in our previous campaigns conducted in Xi'an, Baoji, Shijiazhuang and Beijing(b); the van Krevelen (VK) diagram of the BBOA and OOA-BB factors resolved in the winter of 2018 in Xi'an (c); and the effects of Ox and BBOA concentrations on the OOA-BB formation (d).

The scatterplot of $f_{44}$ vs. $f_{60}$ for ambient data was also applied to further investigate OA transformation during different periods. As shown in Fig.4a, data during the summer of 2019 in Xi'an were mainly located on the left side ($f_{60}$ = 0.1-0.5%), consistent with the negligible biomass burning influence and

BBOA-absent OA sources in the summer campaign (Duan et al., 2021). In the winter campaign, the data were mainly located in the lower right part with $f_{60}$ ranging from 0.4%-1.4% during reference days, suggesting significant influence of BBOA. The data during SIA-P1 were also mainly located in the lower right part with $f_{60}$ ranging from 0.7%-1.4%, suggesting BBOA also had significant influence during this period. Meanwhile, more data were located in the upper range with higher $f_{44}$ than those in

reference days, suggesting the increased OA aging and secondary formation during SIA_P1. As for SIA_P2, more data were located on the left side with no-BB influence, and the range of $f_{44}$ was further higher than those in reference days and SIA_P1, suggesting that the BBOA influence decreased while the SOA influence and OA oxidation state increased during the SIA_P2. Consistently, from reference days and SIA_P1 to SIA_P2, the contribution of BBOA decreased from 13% and 14% to 6%, and the

OOA-BB contribution decreased from 31% and 22% to 16%, respectively, and the aq-OOA contribution increased largely from 19% and 39% to 61%. The scatterplot of $f_{44}$ vs. $f_{43}$ was also discussed in Fig. 4b in order to study the evolution of SOA. The data points substantially fell into the triangle space derived by Ng et al. (2010), in which higher $f_{44}$ and lower $f_{43}$ are characteristics of more oxidized and aged aerosol, while lower $f_{44}$ and higher $f_{43}$ values represent less oxidized and fresh organics. From reference

days to SIA_P1 and SIA_P2, OA showed the evolution trends moving from the lower right to the upper left in the triangle, suggesting the increased oxidation of OA during SIA-enhanced periods (Ng et al., 2010). Consistently, the POA factors (HOA, COA, CCOA, and BBOA) were concentrated in the bottom of the triangle, while OOA-BB was in an intermediate location, and aq-OOA was at the top left of the triangle with the highest oxidation state.

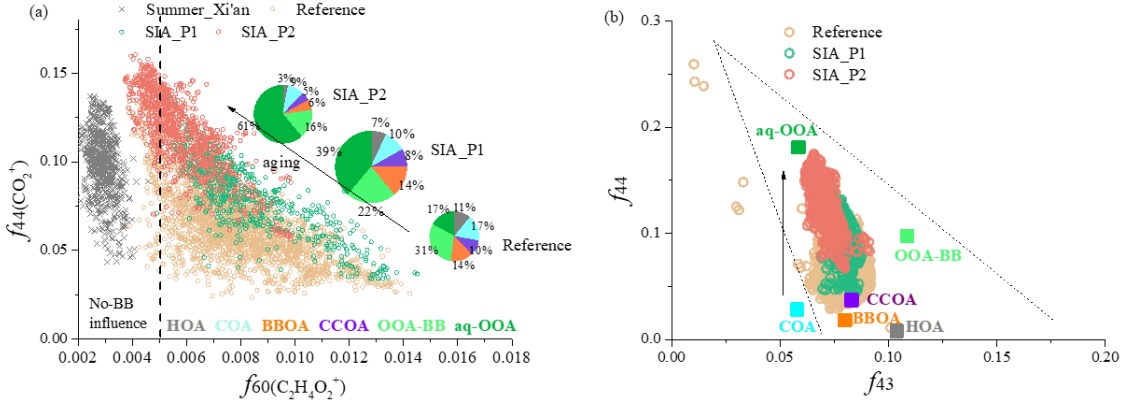


**Fig. 4** The plots of $f_{44}$ vs $f_{60}$, as well as the average OA composition during these three periods (a). The size of the pie chart identifies the mass concentration of total OA, and the plots of $f_{44}$ vs $f_{60}$ in summer 2019 was also shown for comparison. And the scatterplots of $f_{44}$ vs $f_{43}$ (b). The corresponding values of the six OA factors identified in this study are also shown, and the triangle range is from Ng et al. (2010).

**3.3 aq-OOA dependence on SIA and ALWC**

The aq-OOA showed an obvious mass increase during the SIA-enhanced periods, and tracked well with the ALWC increase during this winter campaign (Fig. 1 and Fig. 2). In addition, the mass spectrum of aq-OOA resolved in this study was tightly correlated with that resolved in the summer of 2019 in Xi'an

(Duan et al., 2021) ($R^2 = 0.86$, Fig. S8), and the time series of aq-OOA was also correlated well with

$CH_2O_2^+$ ($R^2 = 0.91$), $CH_3SO^+$ ($R^2 = 0.89$), and $CH_3SO_2^+$ ($R^2 = 0.75$) (Fig. S8), which are the typical fragment ions of aqueous-phase processing products (Tan et al., 2009; Chhabra et al., 2010; Ge et al., 2012; Sun et al., 2016). These results suggested the dominant role of aqueous-phase processes in the formation of aq-OOA in winter Xi'an. As shown in Fig. 5, there was a positive correlation between the concentration of aq-OOA and ALWC with variable slopes in different RH ranges, likely due to the

exponential increase of ALWC with RH (Wu et al., 2018). As discussed by Wu et al. (2018), simultaneously elevated RH levels and SIA concentrations resulted in an abundant ALWC. Condensed water also facilitates the partitioning of water-soluble, polar organics into condensed phases, and subsequent facilitate the SOA formation. Consistently, higher SIA concentration also showed positive effect on the aq-OOA increase (Fig. 5), and a tight correlation between the concentration of aq-OOA

and SIA was observed for the whole dataset irrespective of the RH variation ($R^2 = 0.96$, Fig. S10).

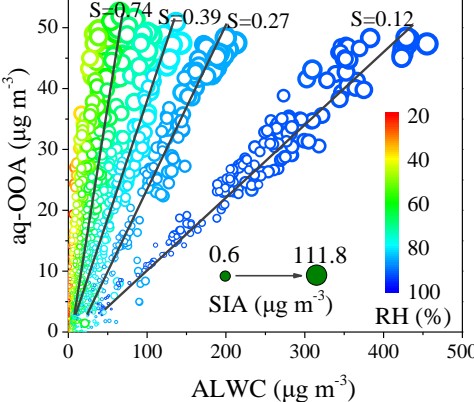

**Fig. 5** The effects of ALWC on the formation of aq-OOA colored by RH, with the increase of SIA concentration shown as the size increase of the data points. Note "S" is defined as the slope between aq-OOA and ALWC in different RH range.

We further compared the aqueous-phase formation of aq-OOA during summer 2019 and winter 2018 in Xi'an, and discussed the specific effect of sulfate or nitrate on their formations (Fig. 6). As discussed in our previous study (Duan et al., 2021), aq-OOA was dominantly formed in fog-rain days with consistently high RH (>60%) and ALWC conditions during summer. The concentration of aq-OOA continuously increased as RH increased from 70% to 100% and ALWC increased from 10 μg m$^{-3}$ to

100 μg m$^{-3}$ and further to 1000 μg m$^{-3}$, suggesting the much important effects of high RH and ALWC on the aq-OOA formation in summer Xi'an (Fig. 6a, b). In addition, nitrate displayed a more positive effect on the aq-OOA formation than sulfate, as sulfate showed a weak correlation ($R^2 = 0.44$) with aq-OOA than that of nitrate ($R^2 = 0.98$) (Fig. S11). Different from aq-OOA in summer that was mainly formed when RH >60%, the formation of aq-OOA in winter was frequently observed when RH >40%

(Fig. 6c, d). This may be related to the much higher nitrate contribution during winter which reduced the deliquesce RH of the aerosol mixture and provided liquid condition for aq-OOA formation at even lower RH (Xue et al., 2014; Wu et al., 2018). When the ALWC was higher than 10 μg m$^{-3}$, aq-OOA was

formed efficiently, and both nitrate and sulfate displayed positive effects on aq-OOA increase. Different from summer, the concentration of aq-OOA was not continuously increasing when ALWC increased from 10-100 µg m⁻³ to >100 µg m⁻³. Instead, the aq-OOA concentration was much affected by the mass increase of nitrate and sulfate, with similar aq-OOA concentration associated with similar sulfate or nitrate concentration level under different RH ranges. This may suggest that aq-OOA formation is more driven by heterogeneous surface reactions in winter, as sulfate and nitrate associated with condensed water may provide the relevant media and increase the aerosol surface area, which leads to the increasing of heterogeneous reaction rate and modulate the formation of SOA (Wu et al., 2018). Also as shown in Fig. 5, the correlation slope (S) between aq-OOA and ALWC decreased from 0.74 for RH <70% to 0.12 for RH >90%, which means that when the ALWC exponentially increased with high RH, aq-OOA did not increase proportionally, and the slope decreased. In comparison, similar aq-OOA concentration was associated with similar SIA concentration levels under different RH ranges. These results suggest that SIA may play a much more important role in the formation of aq-OOA in winter Xi'an.

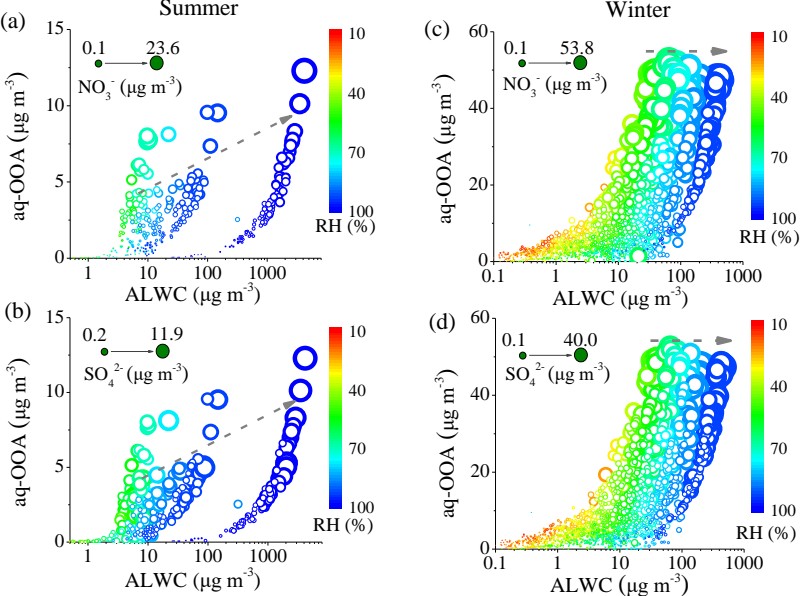

**Fig. 6** Correlations between ALWC and aq-OOA colored by RH during summer (a, b) and winter (c, d) in Xi'an. The effects of nitrate and sulfate are also shown in (a, c) and (b, d), respectively, in which the increase of sulfate or nitrate concentration is shown as the size increase of the data points. The summer data was from Duan et al. (2021), and the horizontal axes both in summer and winter were shown in exponential type for comparison.

### 3.4 Van Krevelen analysis: importance of aqueous-phase processes

The VK diagram, displaying the variation of O/C vs H/C (Hu et al., 2013), was further used to probe OA oxidation reaction mechanisms in our study. As shown in Fig. 7a, data with a higher O/C ratio and lower H/C ratio located in the right-bottom corner were usually related to higher SIA concentration, and higher ALWC also facilitated the increase of O/C ratio, suggesting the positive effects of SIA and

aqueous-phase processes on the OA oxidation enhancement during winter Xi'an. The slope and intercept of the VK diagram for OA during different periods were further displayed in Fig. 7b. More data were located in the right-bottom corner with a higher O/C ratio during SIA-enhanced periods than those during reference days, especially in SIA_P2 with a much higher fraction of aq-OOA. Meanwhile, the slope of the correlation between H/C and O/C during SIA-enhanced periods was also shallower than that during reference days, which changed from -0.49 during reference days to -0.39 during SIA_P1 and -0.33 during SIA_P2. This variation might suggest the transformation of OA from reference days to SIA-enhanced periods, which is likely transferring much close to the processes of addition of alcohol or peroxide groups (slope ≈ 0) (Heald et al., 2010; Chen et al., 2015).

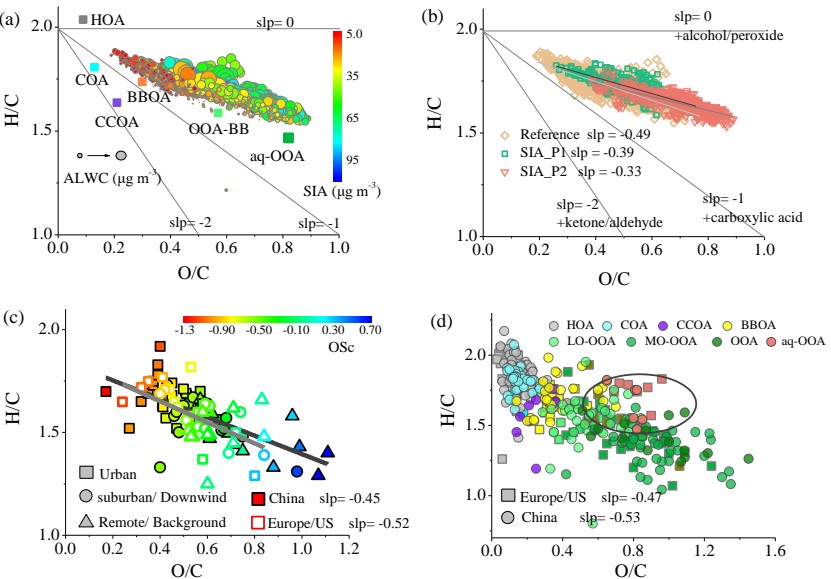

**Fig. 7** The VK diagram of H/C vs O/C for the entire winter observation (a) as well as different periods including reference days, SIA_P1 and SIA_P2 (b). The scatterplots of H/C vs O/C are colored by the mass concentration of SIA, and the size of the data points is proportional to the ALWC concentration in figure a. The scatterplots of H/C vs O/ C of the bulk OA (c) as well as different OA factors (d) observed and resolved in urban, rural and remote sites in recent years both in China and Europe or US based on HR-AMS are also summarized for comparison.

We also explored the oxidation state of bulk OA observed in recent years in China, and compared to those observed in European or American campaigns (Fig. 7c and Table S2). Campaign-averaged O/C ratios of total OA observed in China range from 0.17 to 1.11 with the carbon oxidation states (OSc) ranging from -1.36 to 0.85, which are much variable than those observed in European or American campaigns with the average O/C ratio ranging from 0.24 to 0.84 and OSc ranging from -1.17 to 0.31, respectively. As for different campaigns in China, the O/C ratio of bulk OA increases from an average of 0.45 in urban sites to 0.58 in suburban or rural sites and further to 0.80 in remote or background sites, likely due to the OA aging and less influence from POA emission in rural or background sites. Meanwhile, the slope of H/C vs. O/C for bulk OA observed in China is -0.45, slightly flatter than that

observed in European or American campaigns (-0.52), while both are close to -0.5. This suggests that the carboxylic acid with fragmentation dominates OA aging from urban to remote sites both in China and European or American sites. Similarly, the VK diagram between O/C and H/C ratios for PMF-resolved OA factors from AMS measurements in China is also summarized and compared to those in European or American sites (Fig. 7d). Although O/C ratios of bulk OA between China and Europe display different variation ranges, O/C and H/C ratios of specific OA factors show similar range between China and Europe or the US. Similar slope for OA factors evolution from POA to SOA is observed between China and Europe or US, which indicates the consistent characteristic of individual OA factors resolved using PMF. MO-OOA shows the highest O/C ratio, with a range of 0.58-1.35 in China and 0.63-1.49 in Europe or US, respectively. In comparison, aq-OOA is mainly located in the range with both high O/C ratio (0.7-1.0) and high H/C ratio ($\geq$ 1.45), which may result in a much shallower slope for the evolution from POA to SOA (Heald et al., 2010; Chen et al., 2015). This was also consistent with the slope changes from reference days to SIA-enhanced periods for bulk OA observed in our study, as the aq-OOA enhanced obviously during the SIA-enhanced periods.

## 4 Conclusion

The NR-PM$_{2.5}$ chemical composition and OA sources were characterized during the heating season of 2018 in Xi'an. The average mass concentration of NR-PM$_{2.5}$ was 68.0 $\pm$ 42.8 µg m$^{-3}$, higher than that during the summer of 2019 but much lower than that during the winter of 2013 in Xi'an. Six OA sources including HOA, COA, CCOA, BBOA, OOA-BB, and aq-OOA were resolved, in which SOA contributed a much larger extent (58%) than POA (42%) to total OA mass. Further formation mechanism analysis showed that OOA-BB was mainly formed from the photochemical oxidation and aging of BBOA, which formation was more favorable in the reference days with higher BBOA concentration. In comparison, aq-OOA was dominated by the aqueous-phase processes, which showed an obvious mass increase during the SIA-enhanced periods, and tracked well with the ALWC. From reference days to SIA-enhanced periods which usually related to haze pollution, aq-OOA increased obviously, with the concentration and fraction increasing from 4.9 $\pm$ 3.7 µg m$^{-3}$ (17%) during reference days to 26.2 $\pm$ 14.6 µg m$^{-3}$ (39%) during SIA_P1 and 22.7 $\pm$ 10.7 µg m$^{-3}$ (61%) during SIA_P2, respectively, suggesting the critical role of aqueous-phase processes in haze pollution during winter in Xi'an. Consistently, the O/C ratio of the bulk OA increased from 0.41 during reference days to 0.52 during SIA_P1 and 0.67 during SIA_P2, with the VK slope of H/C vs O/C changing from -0.49 to -0.39 and -0.33, respectively. This suggests the increased aq-OOA contribution during SIA-enhanced periods is likely transferring the OA evolution close to the processes of addition of alcohol or peroxide groups. The comparison of oxidation state of bulk OA or OA factors observed in recent years further indicates that carboxylic acid with fragmentation dominates OA aging from urban to remote sites both in China and European or American sites. Meanwhile, aq-OOA mainly located in the range with both high O/C ratio and high H/C ratio might also result in a much shallower slope close to the alcohol or peroxide addition in the OA oxidation processes from POA to SOA.

*Data availability.* Raw data used in this study are archived at the East Asian Paleoenvironmental Science Database, National Earth System Science Data Center, National Science & Technology Infrastructure of China (http://paleodata.ieecas.cn/index.aspx).

*Supplement.* The Supplement related to this article is available online at

475 *Author contributions.* RJH designed the study. JD, YFG, CSL, and HBZ conducted the field observation. Data analysis and source apportionment were done by JD and RJH, with help from WX, QL, and YY. JD and RJH wrote the manuscript. JD and RJH interpreted data and prepared display items, and JO, DC, TH, and CO all commented on and discussed the manuscript.

*Competing interests.* The authors declare that they have no conflict of interest.

**Acknowledgements.** This work was supported by the National Natural Science Foundation of China (NSFC) under grant no. 41925015, the Strategic Priority Research Program of Chinese Academy of Sciences (No. XDB40000000), the Chinese Academy of Sciences (no. ZDBS-LY-DQC001), and the Cross Innovative Team fund from the State Key Laboratory of Loess and Quaternary Geology (SKLLQG) (no. SKLLQGTD1801).

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
