# Peer review of "Measurement report: Large contribution of biomass burning and aqueous-phase processes to the wintertime secondary organic aerosol formation in Xi'an, Northwest China"

_Atmospheric Chemistry and Physics, 2022_

## Author Comment (AC1)

**Response to Referee #1**

General comments:

This paper uses an SP-L-ToF-AMS to study the OA sources and SOA formation in urban Xi'an during winter 2018. The authors used AMS source apportionment techniques to study the contribution of different types/sources to OA and performed correlational analysis to identify key factors on the observed trends of different OA. They further focus on OOA derived from biomass burning (OOA-BB) and aq-OOA from aqueous reactions. In particular, aq-OOA was found to be dependent on SIA content and ALWC. The use of AMS type instruments for source apportionment analysis and the identification of SOA is quite routine now. POAs (HOA, COA, CCOA, BBOA etc) have been regularly identified. The observations of different types of OOAs, their VK plot characteristics and OSc trends are also widely reported. While the paper is a robust AMS "Measurement Report", I am most interested in the interplay between sulfate and nitrate and SOA formation, especially under different environments, BBOA dominant vs. aqueous phase chemistry dominant. Below I highlight a few comments for the consideration of the authors.

**Response:** We thank the referee to review our manuscript and particularly for the valuable comments and suggestions that are very helpful in improving the manuscript. We agree with the referee that the use of AMS type instruments for source apportionment analysis and the identification of SOA is routine now. However, multiple control measures have been implemented in Xi'an in recent years and aerosol composition is expected to have large variations, while direct elucidation and characterization are lack. In our study, PM2.5 composition was measured during the heating season of 2018 in Xi'an using a long-time-offlight AMS (LToF-AMS). Chemical composition and OA sources were analyzed and compared with those resolved in 2013 winter in Xi'an (Elser et al., 2016), in order to elucidate the aerosol variation in recent years due to emission controls. Our results indicated that the formation of OOA-BB was more favorable in the days with larger OA fraction and higher BBOA concentration. In comparison, aq-OOA was more dependent on SIA and ALWC. In particularly, the concentration of aq-OOA was not continuously increasing when ALWC increased from 10- $100 \ \mu g \ m^{-3}$  to >100  $\ \mu g \ m^{-3}$ . Instead, the aq-OOA concentration was much affected by the mass increase of nitrate and sulfate, with similar aq-OOA concentration associated with similar sulfate or nitrate concentration level under different RH ranges, suggesting that SIA may play a much important role in the formation of aq-OOA in winter Xi'an. These results will further complete our understanding of SOA formation. We provide below point-by-point responses to the referee's comments. We also have made most of the changes suggested by the referee in the revised manuscript.

Page 4 line 135, what is the justification of using C2H3O+ as an identifier of OOA-BB? Also, the close relationship between BBOA and OOA-BB without much time lag can also mean that the OOA-BB measured could be a result of oxidation occurring very close to the source of the

burning instead of the result of atmospheric aging under "environmental conditions". Can the authors examine their data in more details or literature to address this issue? My main concern is that atmospheric aging does take time and a time lag is expected.

**Response:** We thank the referee's comment. m/z 43 (mainly C2H3O+) is an indicator of less oxidized oxygenated organics, while m/z 44 (mainly CO2+) is an indicator of more oxidized oxygenated organics (Canonaco et al., 2015; Wang et al., 2017). The tight correlation between the time series of C2H3O+ and OOA-BB indicate its less oxidized property. And its source influence from biomass burning was further identified by the correlation between OOA-BB and BBOA as well as the formation mechanism discussed in section 3.2.

Meanwhile, we have also examined our data in more details, and found that although moderate correlation was observed between the time series of OOA-BB and BBOA ( $R^2=0.59$ ), lags and differences between their time series were observed for certain periods (Fig. R1), as expected by the referee. However, as the chemical processes and atmospheric aging are influenced by many parameters such as meteorology, Ox concentration, solar radiation, the time lag between OOA-BB and BBOA was not consistent and showed variations over the entire measurement period.

In the revised manuscript page 10 lines 303-305, we have now added the sentence ".....Note that although moderate correlation was observed between the time series of OOA-BB and BBOA, lags and differences between their time series were observed, suggesting the atmospheric aging under environmental conditions".

Fig. R1 Time series of BBOA and OOA-BB during the measurement.

Page 7 line 213, did the SOA concentration increase? An increase in SOA% could be due to lower POA.

**Response:** We thank the referee for pointing this out. We checked the data and found that the average SOA concentration observed in our measurement (21.8  $\mu$ g m-3) was higher than that observed during reference days (5.4  $\mu$ g m-3), while lower than that during extreme haze (47.0  $\mu$ g m-3) in the winter of 2013 in Xi'an (Elser et al., 2016). To be more accurate, in the revised manuscript page 9 lines 276-278, we have now deleted ".....suggesting enhanced formation of SOA in recent years", and the sentence now reads "The contribution of SOA was much higher than that observed in the winter of 2013 in Xi'an (16%, 5.4 ± 8.9  $\mu$ g m-3 in reference days and 31%, 47.0 ± 12.0  $\mu$ g m-3 in haze days, respectively) (Elser et al., 2016)".

Page 7 line 221, what is the typical time scale for such BBOA reactions to form OOABB? Is it reflected in the correlation of time-lag profile of OOA-BB with time profile of BBOA? **Response:** We thank the referee's comment. In response above, time lag and difference between the time series of OOA-BB and BBOA were observed (Fig. R1). However, as the chemical processes and atmospheric aging are influenced by many parameters such as meteorology, Ox concentration, solar radiation, the time lag between OOA-BB and BBOA was not consistent and showed variations over the measurement.

Page 10 Fig.4, why was the slope larger at low RH? It is interesting and the authors should explain. The size of the symbols is meant to show the SIA concentration. But what is the scale? **Response:** We thank the referee's suggestion. As we discussed on page 14 lines 431-439 in the revised manuscript, the correlation slope (S) between aq-OOA and ALWC decreased from 0.74 for RH <70% to 0.12 for RH >90%, which means that when the ALWC exponentially increased with high RH, aq-OOA did not increase proportionally, and the slope decreased. In comparison, similar aq-OOA concentration was associated with similar SIA concentration levels under different RH ranges. These results suggest that SIA may play a much more important role in the formation of aq-OOA in winter Xi'an.

In addition, we have now added the scale of symbols to show the SIA concentration in Fig. 5 in the revised manuscript (see Fig. R2 below). Note the figure has also been updated according to suggestions from the other referee.

Fig. R2 The effects of ALWC on the formation of aq-OOA colored by RH, with the increase of SIA concentration shown as the size increase of the data points. Note "S" is defined as the slope between aq-OOA and ALWC in different RH ranges.

Page 10, line 295, as sulfate and nitrate increased, the aqOOA also increased to the same high level at some different ALWs. So, within certain range of ALW, its increase correlates with aqOOA increase. But at a very large increase of ALW, it does not give much higher max of aqOOA. What are the reasons for this?

**Response:** We thank the referee's comment. Condensed water would facilitate the partitioning of water-soluble, polar organics into condensed phases, and subsequent aqueous- phase SOA formation (Wu et al., 2018), thus within a certain range of ALWC, its increase correlates with aq-OOA increase. However, as we further discussed in Figs. 5 and 6 in the revised manuscript, the concentration of aq-OOA was not continuously increasing with a large increase of ALWC from 10-100  $\mu$ g m-3 to >100  $\mu$ g m-3. Instead, the aq-OOA concentration was much affected by the mass increase of nitrate and sulfate, with similar aq-OOA concentrations associated with similar sulfate or nitrate concentration level under different RH ranges. This further suggested that SIA may play a much more important role in the formation of aq-OOA in winter Xi'an.

Page 11 Fig. 5, the summer data are from Duan et al., (2021) should be stated in the captions. Also in Fig. 5, what are the scales for nitrate and sulfate concentrations?

**Response:** Change made. In the revised manuscript, the summer data from Duan et al. (2021) has been stated in the caption of the figure, and the scales of symbols to show the sulfate and nitrate concentrations are also added (see Fig. R3 below).

---

## Editor Decision (ED1)

l. 31: 'carboxylic acid with fragmentation' is not clear. Do you mean 'carboxylic acid formation due to fragmentation of larger precursors' or 'fragmentation of carboxylic acids'? Please clarify.

l. 32: 'was likely transferring the OA evolution close to the processes of addition of alcohol or peroxide groups' is not clear. Do you mean 'likely reflect OA evolution due to the addition of alcohol or peroxide groups'? Please clarify.

l. 53: replace 'cognition' by 'knowledge'

l. 60: add 'is' (which is defined as...)

l. 79: either remove 'are' or replace 'lack' by 'lacking'.

l. 83: replace 'absorption' by 'absorbing'

l. 89/90: replace 'much consistent with' by 'very similar to'

l. 117: replace 'gases species' by 'gas species', or remove 'species'

l. 125: Please add reference for the SQUIRREL and PIKA software. To find appropriate citations, please look at https://cires1.colorado.edu/jimenez-group/wiki/index.php/ToF-AMS_Analysis_Software

l. 1321: Strictly OM/OC is not an elemental ratio. Better write 'The elemental ratios including oxygen-to-carbon (O/C), and hydrogen-to-carbon (H/C), as well as the organic mass-to-organic carbon (OM/OC) ratio were also analyzed...'

l. 150: replace 'increase of' by 'higher'

l. 163: replace 'transformation of nitrate from gas-phase to particle-phase' by 'partitioning of nitrate to the particle phase'

l. 201: replace 'much' by 'very'

l. 217: 'the increase ratio of sulfate or nitrate contribution from reference days to SIA periods' is not fully clear. It might be better to write 'the increased contribution of sulfate or nitrate during SIA periods as compared to reference days ('increase ratio', IR)'

l. 239: add 'more' before 'favorable'

l. 243: replace 'meteorology' by 'meteorological'

l. 244: remove 'species'

l. 303: It is not clear what you mean by 'group papers'. Certainly, f44 and f60 were measured by other groups before – perhaps not at the same location. What do you mean here? Would it be sufficient to shorten the text to '(see Fig. 3b, additional data from our previous studies are also shown)'?

l. 317: 'positively' seems redundant here and could be removed

l. 317, 320: 'Meanwhile' is redundant in both sentences

l. 323: 'Plots of' is redundant in a figure caption

l. 354: remove 'The plots of'

Figure 4: The numbers next to the pie charts are very small. I suggest moving them to a separate panel (4c) and adding the values of the mass concentrations there. It is sufficient in panel a) to write 'SIA_P2', 'SIA_P1' and 'Reference' when the additional information is then included in a new, clearer panel c.

l. 393: replace 'weak' by 'weaker'

l. 403: specify 'much more important' than what? Than during summer? Or than the increase in RH?

l. 422: clarify 'carboxylic acid with fragmentation (slope=-0.5) dominated OA aging in reference days, and the variation of slope might suggest the transformation of OA from reference days to SIA-enhanced periods, which is likely transferring much close to the processes of addition of alcohol or peroxide groups' (see my comments to your abstract)

l. 452: Data availability statement: Would it be possible for you to extract the data that were used for your paper (and only those) from the large data base, and provided them in a separate repository? If not, it would be at least helpful if you could give a bit more information in the data availability statement how and where to find the data in the large data base.

---

## Author Response (AR2)

**Response to Referee #2**

First of all, I would like to thank the authors for their thorough responses and revisions to the referee comments. Upon reading the manuscript after the authors' substantial improvements to clarity and background, I am able to better understand the very interesting content of the article and the comprehensive scope of the dataset. The main referee suggestion to add additional context and interpretation based on prior literature has been well addressed, as has the concern about proper attribution. While the organization of the paper has improved, I have two additional suggestions for the results and discussion. Further consolidation of the many small figures would enhance the clarity of the story, but I will leave that to the discretion of the authors. I have also added some final comments on the clarity of the added content.

The authors' revisions substantially improve the quality of the paper in terms of the communication of ideas, references to prior work that is relevant, and overall scientific value of the content. I still have some suggestions, but I recommend this paper for publication after minor revisions.

**Response:** We thank the referee to review our manuscript and particularly for the valuable comments and suggestions that are very helpful in improving the manuscript. Below is our point-by-point response to each comment.

Specific Comments

Lines 370-396: I find this section to be the most fascinating in the paper. This work expands upon that of Wu et al., 2018 by applying this scenario to the aq-OOA concentration. However, I have two concerns:

1. I found that the description presented in the authors' response to the referee was more clearly laid out than that described in the article: describing the summertime observations (ALWC dependence) and hypothesis (aq-OOA formation from bulk aqueous reactions), followed by the contrasted wintertime observations (SIA dependence) and hypothesis (heterogeneous surface reactions).

**Response:** We thank the referee's comment. In the revised manuscript page 13 lines 401-435, we have now updated this paragraph according to the description presented in the last authors' response to the referee. It now reads: "……aq-OOA was dominantly formed in fog-rain days with consistently high RH (>60%) and ALWC conditions during summer. The concentration of aq-OOA continuously increased as RH increased from 70% to 100% and ALWC increased from 10 µg m$^{-3}$ to 100 µg m$^{-3}$ and further to 1000 µg m$^{-3}$(Fig. 6a, b). This suggested the formation of aq-OOA was much dependent on ALWC, which might be a bulk water reaction in summer Xi'an (Duan et al., 2021). In comparison, the concentration of aq-OOA was not continuously increasing when ALWC increased from 10-100 µg m$^{-3}$ to >100 µg m$^{-3}$. Instead, the aq-OOA

concentration was much affected by the mass increase of nitrate and sulfate, with similar aq-OOA concentration associated with similar sulfate or nitrate concentration level under different RH ranges (Fig. 6c, d). This may suggest that aq-OOA formation is more driven by heterogeneous surface reactions in winter, as sulfate and nitrate associated with condensed water may provide the relevant media and increase the aerosol surface area, which leads to the increasing of heterogeneous reaction rate and modulate the formation of SOA (Wu et al., 2018). Nitrate displayed a more positive effect on the aq-OOA formation than sulfate in summer, as sulfate showed a weak correlation ($R^2 = 0.44$) with aq-OOA than that of nitrate ($R^2 = 0.98$) (Fig. S11). Different from aq-OOA in summer that was mainly formed when RH >60%, the formation of aq-OOA in winter was frequently observed when RH >40% (Fig. 6c, d). This may be related to the much higher nitrate contribution during winter which reduced the deliquesce RH of the aerosol mixture and provided liquid condition for aq-OOA formation at even lower RH (Xue et al., 2014; Wu et al., 2018). When the ALWC was higher than 10 μg m$^{-3}$, aq-OOA was formed efficiently, and both nitrate and sulfate displayed positive effects on aq-OOA increase. Also as shown in Fig. 5, the correlation slope (S) between aq-OOA and ALWC ……".

2. Even given the clearer explanation, it looks in Figure 6 as though there are simply more data points at higher aq-OOA concentrations in the winter dataset, and, based on the figure and current discussion, I'm not entirely compelled to agree with the authors' hypothesis (that there is a difference in summer versus winter aq-OOA relationships with ALWC and SIA concentrations, respectively). Please directly explain how the dependence on ALWC implies a bulk reaction scheme, while SIA mass dependence implies a surface area scheme, and relate that to the data. Including example values and directly discussing the arrows in Figure 6 would also be helpful to support the hypothesis. In addition, please consider the supposition by Wu et al. that wet deposition over 80 % RH impacts the observed relationship. Does this work support that finding?

**Response:** We thank the referee's comment and suggestion. As shown in Figure 6, the trend of the arrows between summer and winter showed that the concentration of aq-OOA continuously increased as ALWC increased from 10 μg m$^{-3}$ to 100 μg m$^{-3}$ and further to 1000 μg m$^{-3}$ in summer. This suggested the formation of aq-OOA was much dependent on ALWC, which might be a bulk water reaction in summer Xi'an. In comparison, the concentration of aq-OOA was not continuously increasing with the increase of ALWC. Instead, the aq-OOA concentration was much affected by the mass increase of nitrate and sulfate, with similar aq-OOA concentration associated with similar sulfate or nitrate concentration level under different RH and ALWC ranges. As discussed in Wu et al. (2018), increased hygroscopic particle constituents such as sulfate and nitrate associated with condensed water may provide the relevant media and increase the aerosol surface area, which leads to the increasing of heterogeneous reaction rate and modulate the formation and properties of SOA. Therefore, we stated here that aq-OOA formation is more driven by heterogeneous surface reactions. We agreed that wet deposition

over 80 % RH might impact the observed relationship (Wu et al., 2018). However, as shown in Figure 6, similar relationship between aq-OOA and ALWC or SIA could be found for data with RH <80% (shown as different color in the figure), in which aq-OOA was still much affected by the mass increase of nitrate and sulfate, with similar aq-OOA concentration associated with similar sulfate or nitrate concentration level under different RH and ALWC ranges.

Van Krevelen discussion (lines 310-312, Section 3.4, Abstract, Conclusion):

1. I agree with the authors that the Van Krevelen analysis is useful to demonstrate the similarity of chemical pathways in different regions/studies (as in Heald et al., 2010) or the chemical pathway in higher resolution data (as in Ng et., 2011). I applaud the very thorough and interesting analysis in Section 3.4, and I agree that discussing which molecular pathways could be at play makes sense with the higher time resolution datasets. However, this expanded analysis does not really fit, and could be a starting point for an entirely new paper in terms of content. Please clarify how the expanded Van Krevelen discussions support the story this paper tells, and add an explanation in the text and/or save some of the analysis for another manuscript. Since a slope over multiple field campaigns and lab studies is already available in the literature (Heald et al., 2010 and possibly others), it might be sufficient for this paper to contrast the current field campaign values with those from the literature.

**Response:** We thank the referee's comment and suggestion. We agree with the referee that analysis of current field campaign values is sufficient and the expanded Van Krevelen discussions over multiple field campaigns could be saved as a starting point for an entirely new paper. According to the referee's suggestion, in the revised manuscript, section 3.4 has been reorganized, in which the paragraph "We also explored the oxidation state of bulk OA observed in recent years in China……This was also consistent with the slope changes from reference days to SIA-enhanced periods for bulk OA observed in our study, as the aq-OOA enhanced obviously during the SIA-enhanced periods" has been removed.

Meanwhile, Figure 7 has been updated as follows:

[Figure]

Figure R1 The VK diagram of H/C vs O/C for the entire winter observation (a) as well as different periods including reference days, SIA_P1 and SIA_P2 (b). The scatterplots of H/C vs O/C are colored by the mass concentration of SIA, and the size of the data points is proportional to the ALWC concentration in figure a.

In addition, in the revised manuscript page 17 lines 508-512, the conclusion of "The comparison of oxidation state of bulk OA or OA factors observed in recent years further indicates that carboxylic acid with fragmentation dominates OA aging from urban to remote sites both in China and European or American sites. Meanwhile, aq-OOA mainly located in the range with both high O/C ratio and high H/C ratio might also result in a much shallower slope close to the alcohol or peroxide addition in the OA oxidation processes from POA to SOA" has also been removed.

In the revised supplement, Table S2 as well as related references have also been removed.

2. The chemical pathways that the authors give do not seem to align with the slopes calculated from the data. For example, the slopes of -0.39 and -0.33 shown in Figure 7b are attributed to addition of alcohol or peroxide groups, which would be a slope of 0 (assuming no fragmentation). Some combination of moieties that could add to the slope that is found (i.e., -0.39 and -0.33) should be given. Please revisit, and perhaps more clearly explain the reasoning, for the slopes in the Conclusion and Section 3.4 in particular.

**Response:** We thank the referee's suggestion. As shown in Figure 7b, the slope of the correlation between H/C and O/C during reference days was -0.49, close to -0.5 which indicates the dominant OA aging processes of the carboxylic acid with fragmentation (Chen et al., 2015). In comparison, the slope changed from -0.49 to -0.39 in SIA_P1, and further to -0.33 in SIA_P2, with the increased aq-OOA contribution. As the slope of -0.39 and -0.33 were shallower and more close to 0 than that of -0.49, thus we supposed that the transformation of OA aging processes from reference days to SIA-enhanced periods, which is likely transferring much close to the processes of addition of alcohol or peroxide groups (slope $\approx 0$) (Heald et al., 2010; Chen et al., 2015). To be more accurate, in the revised manuscript page 15 lines 453-457, the sentence has been updated, which now reads "Meanwhile, the slope of the correlation between H/C and O/C during SIA-enhanced periods was also shallower than that during reference days, which changed from -0.49 during reference days to -0.39 during SIA_P1 and -0.33 during SIA_P2. This suggested the carboxylic acid with fragmentation (slope=-0.5) dominated OA aging in reference days, and the variation of slope might suggest the transformation of OA from reference days to SIA-enhanced periods, which is likely transferring much close to the processes of addition of alcohol or peroxide groups (slope $\approx 0$) (Heald et al., 2010; Chen et al., 2015)".

3. If Figures 7c and 7d are kept, please list the references for all of the studies included as data points (or cite where the references can be found if they are already included as a list elsewhere).

**Response:** We thank the referee's suggestion. As response above, in the revised manuscript, section 3.4 has been reorganized and Figures 7c and 7d have been removed, according to the referee's suggestion.

4. Lines 428 onward: Please describe the difference between "the O/C ratio of bulk OA" (line 430) and "the average O/C ratio" (line 429). The values listed in this paragraph don't seem to match those in Figures 7c and 7d. Please verify that the values in the text are accurate.

While the Conclusions align well with the major findings of the paper, the Abstract is not as representative. Please consider updating the Abstract to better align with the content of the paper.

**Response:** We thank the referee's comment. "The O/C ratio of bulk OA" and "the average O/C ratio" are both referred as the O/C ratio of OA reported in different campaigns in previous studies. As response above, in the revised manuscript, this paragraph as well as Figures 7c and 7d have been removed. In addition, in the revised manuscript, according to the referee's suggestion, the Abstract has been updated to better align with the content of the paper. It now reads "……In comparison, the aqueous-phase processed oxygenated OA (aq-OOA) was more dependent on secondary inorganic aerosol (SIA) content and aerosol liquid water content (ALWC), and increased largely to 50% of OA during SIA-enhanced periods. Further Van Krevelen (VK) diagram analysis suggests the carboxylic acid with fragmentation dominated OA aging in reference days, while the increased aq-OOA contributions during SIA-enhanced periods is likely transferring the OA evolution close to the processes of addition of alcohol or peroxide groups". ".

Please write out the objectives and/or hypotheses of the paper in a statement at the end of the Introduction section. Several sentences throughout the Introduction give hints (lines 74-75 "...found the contribution of SOA increased…", lines 76-77 "...it is expected that…", lines 89-90 "these studies indicated…"), but there should be a clear statement. It seems to me that the two main objectives are along the lines of: (1) finding which variables control aq-OOA formation and how; and (2) quantifying the changing contributions to OOA between seasons and years at Xi'an.

**Response:** We thank the referee's suggestion and change made. In the revised manuscript, a statement for the objectives of the paper has been added at the end of the Introduction section. It now reads "…… and its contribution to haze event were investigated and compared with those in the summer of 2019 (Duan et al., 2021). The main objectives of our study were to investigate the dominant variables mediating aq-OOA formation, and to quantify the changing contributions of SOA between seasons and years at Xi'an".

Please consider moving the paragraph at line 224 ("Six OA sources were resolved…") to before the paragraph at line 166 ("A continuous and large increase…"). The paragraph at line 224 is more related to the theme of changes over time in OOA sources, which is discussed above line 166.

**Response:** We thank the referee's suggestion. In the revised manuscript page 5 lines 171-243,

the paragraph of "Six OA sources were resolved…" has been moved to before the paragraph of "A continuous and large increase…". The paragraph now reads "……Six OA sources were resolved, ……The contribution of SOA was much higher than that observed in the winter of 2013 in Xi'an (16%, 5.4 ± 8.9 μg m$^{-3}$ in reference days and 31%, 47.0 ± 12.0 μg m$^{-3}$ in haze days, respectively) (Elser et al., 2016).

A continuous and large increase of secondary inorganic aerosol……These results suggested that high RH is favorable in sulfate formation than nitrate formation especially in haze pollution in winter Xi'an".

Line 258: Please add takeaways about the paragraph: is the key addition to the paper's story that the SIA_P2 represents a period of greater oxidation?

**Response:** We thank the referee's suggestion and change made. In the revised manuscript page 9 lines 285-286, the sentence now reads "……much higher than those during reference days and SIA_P1. This suggested the much higher OA oxidation state during SIA_P2 than reference days and SIA_P1".

Line 303: "In comparison, the MO-OOA resolved in Baoji…" - should this be the LO-OOA? The LO-OOA looks closer to the biomass burning region in the figure.

**Response:** Change made. The sentence now reads "In comparison, the LO-OOA resolved in Baoji…".

In addressing the referee comment 5a regarding the expectations for the PMF factors, the following background was provided for the referee. Please consider adding this perspective to the section of the manuscript discussing POA factors.

"Agricultural burning was an important contributor to OA in harvest season before 2013. However, agricultural burning in harvest season has been banned after 2013 and BBOA source becomes a negligible contributor to OA in summertime, especially in urban city in recent years (Huang et al., 2021)."

**Response:** We thank the referee's suggestion. In the revised manuscript page 12 lines 353-356, this perspective has been added, which now reads "……consistent with the negligible biomass burning influence and BBOA-absent OA sources in the summer campaign (Duan et al., 2021). Agricultural burning was an important contributor to OA in harvest season before 2013. However, agricultural burning in harvest season has been banned after 2013 and BBOA source becomes a negligible contributor to OA in summertime, especially in urban city in recent years (Huang et al., 2021). In the winter campaign, ……".

Meanwhile, the following reference has been added to the revised reference list:

Huang, L., Zhu, Y., Wang, Q., Zhu, A., Liu, Z., Wang, Y., Allen, D.T., and Li, L.: Assessment of the effects of straw burning bans in China: Emissions, air quality, and health impacts,

Sci. Total. Environ., 789, 147935. https://doi.org/10.1016/j.scitotenv.2021.147935, 2021.

Lines 161-165: Please add concentrations here as well (the authors include contributions) to demonstrate whether concentrations also decreased between Xi'an 2013 and 2018 winters.

**Response:** We thank the referee's suggestion. Concentrations were not discussed here as we focused on the relative contribution and importance of nitrate in total $PM_{2.5}$ between Xi'an 2013 and 2018 winters. The absolute concentration of nitrate in our campaign ($13.3\pm11.4$ µg m$^{-3}$) was similar to that in the reference ($14\pm11$ µg m$^{-3}$), while lower than that in the extreme haze ($71\pm12$µg m$^{-3}$) in Xi'an 2013, due to the extremely high $PM_{2.5}$ concentration ($537\pm146$ µg m$^{-3}$) in haze pollution in Xi'an 2013 winter.

Lines 210-212: Thank you to the authors for clarifying the NOR and SOR relationships. Please also specify what "n" indicates in these equations.

**Response:** Change made. "n" indicates the molar concentration. In the revised manuscript page 7 lines 237-239, the clarification was updated which now reads "……Consistently, although both sulfur oxidation ratio (SOR, defined as $n[SO_4^{2-}]/(n[SO_4^{2-}] + n[SO_2])$, and n indicates the molar concentration of $SO_4^{2-}$ or $SO_2$. Ji et al., 2018; Chang et al., 2020) and nitrogen oxidation ratio (NOR, defined as $n[NO_3^{-}]/(n[NO_3^{-}] + n[NO_2])$, and n indicates the molar concentration of $NO_3^{-}$ or $NO_2$. Ji et al., 2018; Chang et al., 2020) increased with RH……".

Line 88: Is "RH/ALWC" a ratio or specifying that either could be a factor? In Duan et al., 2021, it seems like it is the latter (not a ratio); if that is the case, please use "RH and ALWC" or similar.

**Response:** We thank the referee for pointing this out. "RH/ALWC" is not a ratio, but factors of RH and ALWC. Change made according to the referee's suggestion, and the sentence now reads "……, and Duan et al. (2021) found that persistently high RH and ALWC was the driving factor of aq-OOA formation in summer Xi'an, ……" in the revised manuscript page 3 line 91.

---

## Author Response (AR3)

**Response to Editor**

We thank the editor to review our manuscript and particularly for the valuable comments and suggestions that are very helpful in improving the manuscript. Below is our point-by-point response to each comment.

l. 31: 'carboxylic acid with fragmentation' is not clear. Do you mean 'carboxylic acid formation due to fragmentation of larger precursors' or 'fragmentation of carboxylic acids'? Please clarify.

**Response:** Change made. 'carboxylic acid with fragmentation' means the addition of carboxylic acid groups to the site of a C-C bond cleavage, which results in a VK slope of -0.5, according to Ng et al. (2011). To be more accurate, the sentence has been updated in the revised manuscript, which now reads: "Further Van Krevelen (VK) diagram analysis suggests the addition of carboxylic acid groups with fragmentation dominated OA aging in reference days……".

l. 32: 'was likely transferring the OA evolution close to the processes of addition of alcohol or peroxide groups' is not clear. Do you mean 'likely reflect OA evolution due to the addition of alcohol or peroxide groups'? Please clarify.

**Response:** Change made. In the revised manuscript, this sentence has been updated, which now reads: "……while the increased aq-OOA contributions during SIA-enhanced periods likely reflect OA evolution due to the addition of alcohol or peroxide groups".

l. 53: replace 'cognition' by 'knowledge'

**Response:** Change made.

l. 60: add 'is' (which is defined as...)

**Response:** Change made.

l. 79: either remove 'are' or replace 'lack' by 'lacking'.

**Response:** Change made. 'are' has been removed in the revised manuscript.

l. 83: replace 'absorption' by 'absorbing'

**Response:** Change made.

l. 89/90: replace 'much consistent with' by 'very similar to'

**Response:** Change made.

l. 117: replace 'gases species' by 'gas species', or remove 'species'

**Response:** Change made. 'species' has been removed in the revised manuscript.

l. 125: Please add reference for the SQUIRREL and PIKA software. To find appropriate citations, please look at https://cires1.colorado.edu/jimenez-group/wiki/index.php/ToF-AMS_Analysis_Software

**Response:** We thank the editor's suggestion. The reference for the SQUIRREL and PIKA software has been added in the revised manuscript. The sentence now reads: "The SQUIRREL and PIKA coded in Igor Pro (WaveMetrics) were used to analyze the SP-LToF-AMS data (https://cires1.colorado.edu/jimenez-group/ToFAMSResources/ToFSoftware/index.html, last access: 10 July 2022)".

l. 1321: Strictly OM/OC is not an elemental ratio. Better write 'The elemental ratios including oxygen-to-carbon (O/C), and hydrogen-to-carbon (H/C), as well as the organic mass-to-organic carbon (OM/OC) ratio were also analyzed...'

**Response:** Change made. This sentence has been updated as the editor suggested.

l. 150: replace 'increase of' by 'higher'

**Response:** Change made.

l. 163: replace 'transformation of nitrate from gas-phase to particle-phase' by 'partitioning of nitrate to the particle phase'

**Response:** Change made.

l. 201: replace 'much' by 'very'

**Response:** Change made.

l. 217: 'the increase ratio of sulfate or nitrate contribution from reference days to SIA periods' is not fully clear. It might be better to write 'the increased contribution of sulfate or nitrate during SIA periods as compared to reference days ('increase ratio', IR)'

**Response:** Change made. This sentence has been updated as the editor suggested.

l. 239: add 'more' before 'favorable'

**Response:** Change made.

l. 243: replace 'meteorology' by 'meteorological'

**Response:** Change made.

l. 244: remove 'species'

**Response:** Change made.

l. 303: It is not clear what you mean by 'group papers'. Certainly, f44 and f60 were measured

by other groups before – perhaps not at the same location. What do you mean here? Would it be sufficient to shorten the text to '(see Fig. 3b, additional data from our previous studies are also shown)'?

**Response:** We thank the editor's suggestion. It's sufficient to shorten the text to '(see Fig. 3b, additional data from our previous studies are also shown)', and change made in the revised manuscript.

l. 317: 'positively' seems redundant here and could be removed

**Response:** Change made. 'positively' has been removed in the revised manuscript.

l. 317, 320: 'Meanwhile' is redundant in both sentences

**Response:** Change made. 'Meanwhile' has been removed in the revised manuscript.

l. 323: 'Plots of' is redundant in a figure caption

**Response:** Change made. 'Plots of' has been removed in the revised manuscript.

l. 354: remove 'The plots of'

**Response:** Change made.

Figure 4: The numbers next to the pie charts are very small. I suggest moving them to a separate panel (4c) and adding the values of the mass concentrations there. It is sufficient in panel a) to write 'SIA_P2', 'SIA_P1' and 'Reference' when the additional information is then included in a new, clearer panel c.

Response: We thank the editor's suggestion. In the revised manuscript, Figure 4 has been updated as follows:

[Figure]

Figure R1. $f_{44}$ vs $f_{60}$ during three periods including reference days, SIA_P1, and SIA_P2, $f_{44}$ vs $f_{60}$ in summer 2019 was also shown for comparison (a), $f_{44}$ vs $f_{43}$ during these three periods, the corresponding values of the six OA factors identified in this study are also shown, and the triangle range is from Ng et al. (2010) (b), and the average OA concentration and composition during these three periods (c).

l. 393: replace 'weak' by 'weaker'
**Response:** Change made.

l. 403: specify 'much more important' than what? Than during summer? Or than the increase in RH?
**Response:** We thank the editor's suggestion. The sentence has been updated as: "These results suggest that SIA, compared to RH and ALWC, may play a much more important role in the formation of aq-OOA in winter Xi'an" in the revised manuscript.

l. 422: clarify 'carboxylic acid with fragmentation (slope=-0.5) dominated OA aging in reference days, and the variation of slope might suggest the transformation of OA from reference days to SIA-enhanced periods, which is likely transferring much close to the processes of addition of alcohol or peroxide groups' (see my comments to your abstract)
**Response:** Change made. This sentence has been updated to "This suggested the addition of carboxylic acid groups with fragmentation (slope=-0.5) dominated OA aging in reference days, and the variation of slope might suggest the transformation of OA from reference days to SIA-enhanced periods, which likely reflects OA evolution due to the addition of alcohol or peroxide groups (slope ≈ 0) (Heald et al., 2010; Chen et al., 2015)" according to the editor's suggestion.

l. 452: Data availability statement: Would it be possible for you to extract the data that were used for your paper (and only those) from the large data base, and provided them in a separate repository? If not, it would be at least helpful if you could give a bit more information in the data availability statement how and where to find the data in the large data base.

**Response:** Change made. In the revised manuscript, we have now extracted the data used for our paper from the large data base, and provided them in a separate repository. The data can be accessible by access and download. This section has been updated as follows:

"Data availability statement: The key data sets are archived at the East Asian Paleo environmental Science Database, National Earth System Science Data Center, National Science & Technology Infrastructure of China (http://paleodata.ieecas.cn/FrmDataInfo_EN.aspx?id=635717f4-2b9a-4edb-9a18-e76497ab9925). For further information please contact the corresponding author (rujin.huang@ieecas.cn)".